# Cell stress and phase separation stabilize the monomeric state of pseudoisocyanine chloride employed as a self-assembly crowding sensor
Roland Pollak[1,2,3], Leon Koch [4], Benedikt König[1,3], Sara S. Ribeiro [1,2,3], Nirnay Samanta[2,5], Klaus Huber [4] ✉ & Simon Ebbinghaus [1,2,3] ✉

Cellular stress and ageing involve an increase in crowding and aggregation of amylogenic proteins. We here investigate if crowding is the intrinsic cause of aggregation and utilise a previously established non-protein aggregation sensor, namely pseudoisocyanine chloride (PIC). PIC shows fibrillization in cells into a highly fluorescent J-aggregated state and is sensitive to crowding. Surprisingly, cell stress conditions stabilise the monomeric rather than the aggregated state of PIC both in the cytoplasm and in stress granules. Regarding the different physiochemical changes of the cytoplasm occurring upon cell stress, involving volume reduction, phase separation and solidification, the intrinsic crowding effect is not the key factor to drive associated self-assembly processes.

Amyloid-associated pathologies commonly rely on the unfolding and aggregation of proteins favoured by factors like disease-related mutations or a decrease in solubility induced by cellular aging or stress[1]. Heat stress or the exposure to toxic compounds causes a destabilization of proteins[2,3] leading to denaturation of a small subset of sensitive but functionally important proteins[4]. As a response to such conditions, cells maintain a protein quality control network encompassing hundreds of so-called molecular chaperones[1]. An example is the heat shock protein 70 KDa (Hsp/Hsc70) family of chaperones that shares functions in protein folding, disaggregation, translocation and degradation[5]. Overexpression of Hsp70 mitigates protein aggregation and enhances cell survival[6,7] whereas lowered expression levels trigger protein aggregation and promote cell death[6,8]. However, at elongated stress, these mechanisms are exhausted leading to aggregation that is linked to various neurodegenerative diseases. The connection between protein destabilization and the onset of aggregation is poorly understood[9-13].

Upon stress, physicochemical changes of the cellular milieu are observed. Eukaryotic cells show a transition of the cytoplasm to a more solid-like state with increased mechanical stability and decreased cell volume[14]. This transition is associated with a drop in pH and leads to a decrease in the mobility of organelles and a slow-down of molecular diffusion dynamics as measured by single particle tracking experiments as well as neutron scattering[15-17]. The above-mentioned changes are related to a cellular survival advantage. Di Bari et al. used neutron scattering spectroscopy to investigate the temperature-dependent dynamics of the proteome of *Escherichia coli* (*E. coli*) at nanosecond timescale[15]. They observed a slow-down of cytoplasm dynamics under heat stress occurring close to the cell death temperature. Multiscale molecular dynamics simulations revealed that the slow-down can be attributed to a minor fraction of unfolded protein that forms a sticky network due to increased intermolecular interaction[15].

Another important physicochemical change of the cellular milieu upon stress is the compartmentalization of cells by phase separation. Stress granules, non-membranous compartments in the cytoplasm, are formed as a response to different cellular stresses involving proteins like TDP43 or FUS[18]. Stress granules are capable of recruiting and transiently storing denatured and misfolded proteins[3,18,19]. Embedded in the condensate, the proteins could be shielded from further aggregation events, a mechanism supporting protein quality control[18,20]. On the other hand, non-native proteins could aggregate in stress granules and seed pathological transformations (e.g., liquid-to-solid phase transitions) as observed by a loss in their dynamic nature or a change in morphology[3,18-20]. Thus, it is yet unknown if stress granules support protein quality control mechanisms, or if they are hubs that mainly serve other functions under stress but are conversely prone to malfunction due to the accumulation of misfolded or aggregated proteins.

[1]Lehrstuhl für Biophysikalische Chemie, Ruhr-Universität Bochum, Bochum, Germany. [2]Institut für Physikalische und Theoretische Chemie, TU Braunschweig, Braunschweig, Germany. [3]Research Center Chemical Sciences and Sustainability, Research Alliance Ruhr, Bochum, Germany. [4]Institute of Physical Chemistry, University Paderborn, Paderborn, Germany. [5]Present address: Department of Biochemistry and Molecular Biophysics, Washington University in St Louis, Saint Louis, MO, USA. ✉e-mail: klaus.huber@upb.de; simon.ebbinghaus@rub.de

An intriguing question that we address in this work is whether changes in the crowding effect of the cytoplasm associated with cell stress could promote self-assembly processes, and if the answer is yes, does the partitioning of cells by phase separation mitigates or enhances this effect? Therefore we utilize a previously established self-association sensor based on pseudoisocyanine chloride (PIC)[21–23], which forms fibrillar aggregates by monomeric self-assembly, to specifically analyze the effect of macromolecular crowding at heat stress conditions in the cytoplasm. In comparison to previous approaches that investigated fluorescently tagged aggregation-prone proteins with highly complicated folding and aggregation landscapes[3,19], the idea in this study is to use a simplistic and mostly inert, but kinetically and thermodynamically well-characterized model system, that switches between the monomeric and aggregated state. This allows to analyze the effect of the crowding changes of the cellular milieu on the self-assembly process independently of variable cellular factors like protein quality control mechanisms. PIC self-assembles into fibrillar structures leading to the formation of fibrillar highly fluorescent J-aggregates that can be sensitively detected by live cell imaging[21,22] (Fig. 1a). The monomer shows a weak broad fluorescence peak from 520 to 650 nm with a maximum at around 560 nm (Fig. 1b). In contrast, the fluorescence and absorption of the aggregate shows an intense sharp peak at ~575 nm[24,25] (Figs. 1b and S1). In aqueous solution and 23 °C, a 7 mM PIC solution aggregates reversibly by a decrease in temperature or an increase in concentration[21] (Supplementary Figs. 2 and 3). In crowded environments, attractive crowder interactions increase the aggregation threshold, while macromolecular crowders decrease the threshold due to volume exclusion[21,22]. In crowded HeLa cells, PIC aggregates at a concentration of 50 μM[21].

## Results

### Heat-stressed cells accumulate monomeric PIC

In this study, we first compared the aggregation of PIC in heat-stressed cells forming stress granules to the aggregation of PIC in cells under physiological conditions (Fig. 1a). To induce heat stress, cells were treated at 43 °C for 60 min. The formation of stress granules was detected using a stable HeLa cell line expressing FUS-GFP (HeLa^FUS-GFP). In unstressed cells, FUS-GFP was located in the nucleus of HeLa^FUS-GFP cells engaging in RNA metabolism, transcription or splicing[26]. PIC formed J-aggregates in the cytoplasm of HeLa^FUS-GFP cells (41% ± 5% J-aggregates) within 30 min (Fig. 1a), in accordance with previous studies[21]. J-aggregates were excluded from the nucleus due to their length of about 700 nm assuming a rod-like shape geometry[27]. Monomeric PIC was found in the cytoplasm but also in the nucleus where it can enter due to its small size[28]. In the nucleus, PIC was found to stain nucleoli possibly due to DNA intercalation[29,30]. Surprisingly, in heat-stressed HeLa^FUS-GFP cells, PIC remained fully monomeric (0.2% ± 0.1% J-aggregates in the cytoplasm). We also found that PIC was in FUS-GFP stress granules (Fig. 1a and Supplementary Table 1). PIC fluorescence intensity of the monomer in stress granules was increased compared to the cytoplasm, which could be attributed to RNA or protein interactions (Supplementary Fig. 4). To investigate if applying PIC to cells affected stress granule formation, we characterized stress granules in the presence and absence of PIC. Fluorescence recovery after photobleaching experiments (FRAP) of FUS-GFP in the presence of PIC showed a decrease in the mobile fraction, as well as stress granules of smaller sizes and lower circularities (Fig. 1c–f). This indicates that PIC inclusion in stress granules could change

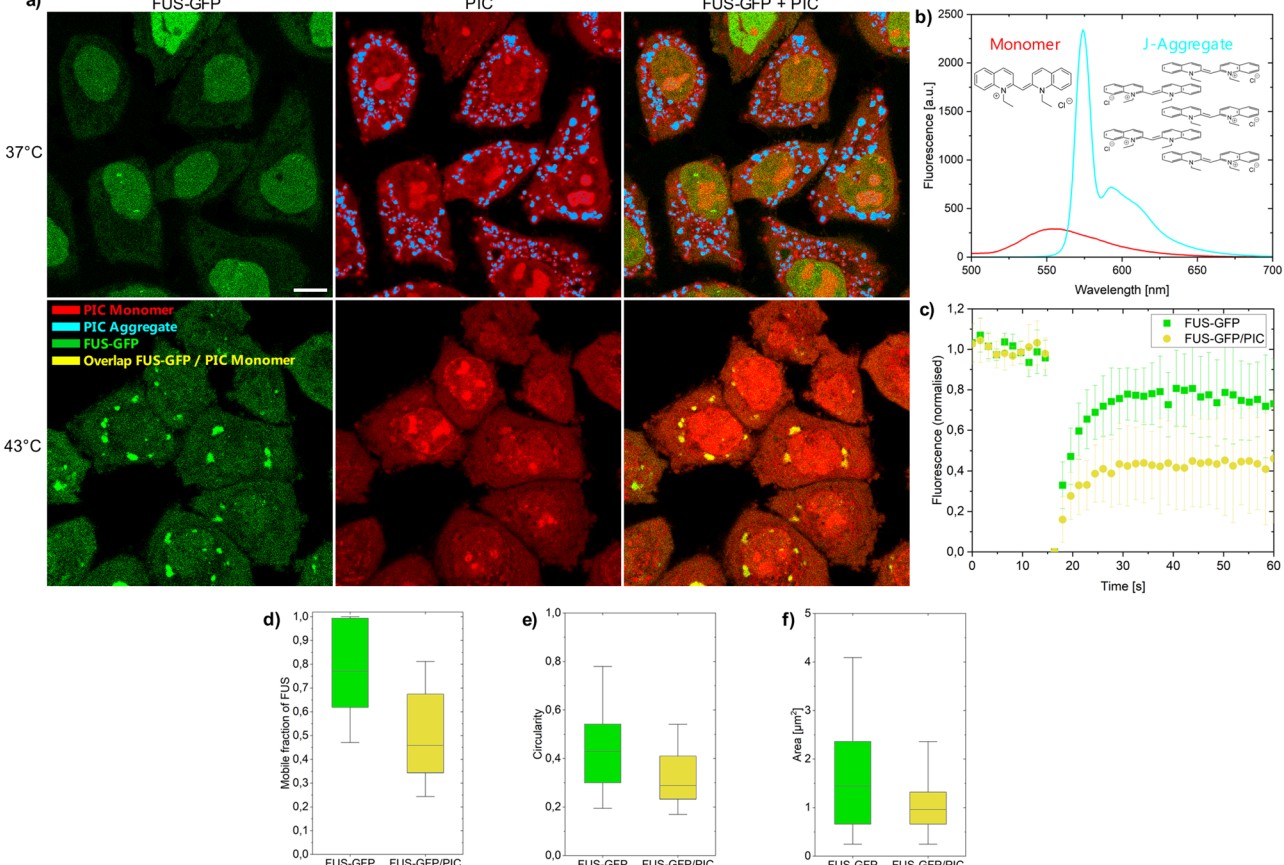

**Fig. 1 | PIC accumulation in HeLa^FUS-GFP cells. a** HeLa^FUS-GFP treated with PIC at 37 °C (upper images) and 43 °C (lower images). At 37 °C, FUS-GFP was mainly located in the nucleus and PIC aggregated in the cytoplasm. At 43 °C FUS-GFP accumulated in the cytoplasm and enriched in stress granules. PIC is monomeric in the cytoplasm and stress granules. J-aggregates (blue) and monomers (red) were assigned based on their intensity and spectrum (**b**) (see "Methods" section for details). Scale bar: 10 μm. **c, d** FRAP measurements of FUS-GFP in stress granules in presence and absence of PIC. **e, f** Circularity and area of stress granules in presence and absence of PIC. **d–f** Boxplot including the first quartile, median and third quartile of 346 individual cells. Whiskers represent 5–95% of all data points. Details in Supplementary Tables 1 and 2.

their architecture, alternatively PIC application during heat stress could induce additional cell stress[23,31] that leads to a different constitution of stress granules.

## Aggregated PIC is excluded from FUS droplets

To further investigate the role of stress granules in the heat stress response, we reconstituted droplets in vitro using the FUS protein. Droplet formation was triggered by TEV cleavage of an MBP-tag from FUS-MBP in solutions of PIC with different concentrations. PIC aggregated in FUS droplets at a solution concentration of 500 μM (Fig. 2a, b), near the aggregation threshold concentration in buffer (>100 μM, Supplementary Fig. 5). The addition of preformed PIC aggregates after FUS droplet formation showed that aggregates enriched on the surface of the droplets (Supplementary Fig. 6) demonstrating that aggregates could not be accommodated in the pre-formed droplets. We further tested heat stress conditions (incubation at 43 °C for 60 min) at which condensates were found to mature into smaller, less circular droplets (Supplementary Fig. 7)[19] and found that PIC formed J-aggregates at 1000 μM (Fig. 2a, c). Comparing PIC-free and PIC-treated FUS droplets, we found only minor morphological changes in size and circularity. To further confirm that PIC does not significantly modify condensates, time-resolved (tr) static and dynamic light scattering (SLS and DLS) measurements were carried out (Fig. 2d). The experiments showed that the addition of PIC did not have a significant effect on the induction period and on the growth phase of FUS droplets. Neither the induction time nor the values of $M_w$, $R_g$ and $R_h$ (see Supplementary Fig. 8 for $R_g$ and $R_h$ and Supplementary Table 5 for $R_g/R_h$) were significantly affected.

## Monomer stabilization under different stress conditions

We then tested sodium arsenate as an alternative cause of cellular stress (arsenate stress)[3,32]. Similar to heat stress, we found no PIC aggregation (0.8% ± 0.4% J-aggregates) under arsenate stress but rather monomeric PIC (Supplementary Fig. 9). We conclude that stress conditions, independent of the stress applied, lead to a change in the crowding of the cytoplasm stabilizing monomeric PIC rather than the J-aggregate.

We then investigated the persistence of the crowding changes for different amplitudes (durations) of heat stress, 30, 60, 90 and 120 min (Supplementary Fig. 10 and Supplementary Table 4). After heat stress, cells were released and incubated for 30 min at 37 °C with PIC added to the medium. The area occupied by J-aggregates in the cytoplasm was 3% ± 1.5% after 30 min of heat stress, 6% ± 3% after 60 min, 0.7% ± 0.7% after 90 min and 0.5% ± 0.7% after 120 min, compared to 41% ± 5% in the non-stressed condition. This showed that prolonged heat stress enhances the stabilization of monomeric PIC, an effect prevailing at least 30 min after recovery to physiological condition.

## Overcrowding suppresses PIC aggregate formation

We tested the effect of increasing crowding conditions using Ficoll 70 as a crowding agent that can be studied to a crowding fraction up to ~50 wt%. Further, it is a polysucrose that is internally cross-linked with epi-chlorohydrin, which is often approximated as a prototypical excluded-volume crowder[33,34]. Indeed, with an increased amount of crowding beyond 10 wt%, the number of micrometer-sized aggregates decreased leading to an almost complete repression at 50 wt% crowder content (Fig. 3a). Further, the threshold temperature for the formation of PIC J-aggregates increased with the concentration of Ficoll 70 up to 2.5 wt% but then dropped and was even lower than that of pure PIC solution at 22.5 wt% (Fig. 3b). This shows parallels with the increase of the excluded volume or reduction of physical space in Ficoll 70 solutions up to 2.5 wt%[35]. However, at concentrations of >2.5 wt% it is assumed that Ficoll 70 molecules undergo compression[35] due to (over-)crowding and further at 10–15 wt% reaching an overlap concentration, where individual Ficoll 70 molecules are forming a transient network[34,36]. This change in crowding environment is followed by stabilizing and destabilizing effects on the crowding sensor PIC as well as protein self-assembly[37].

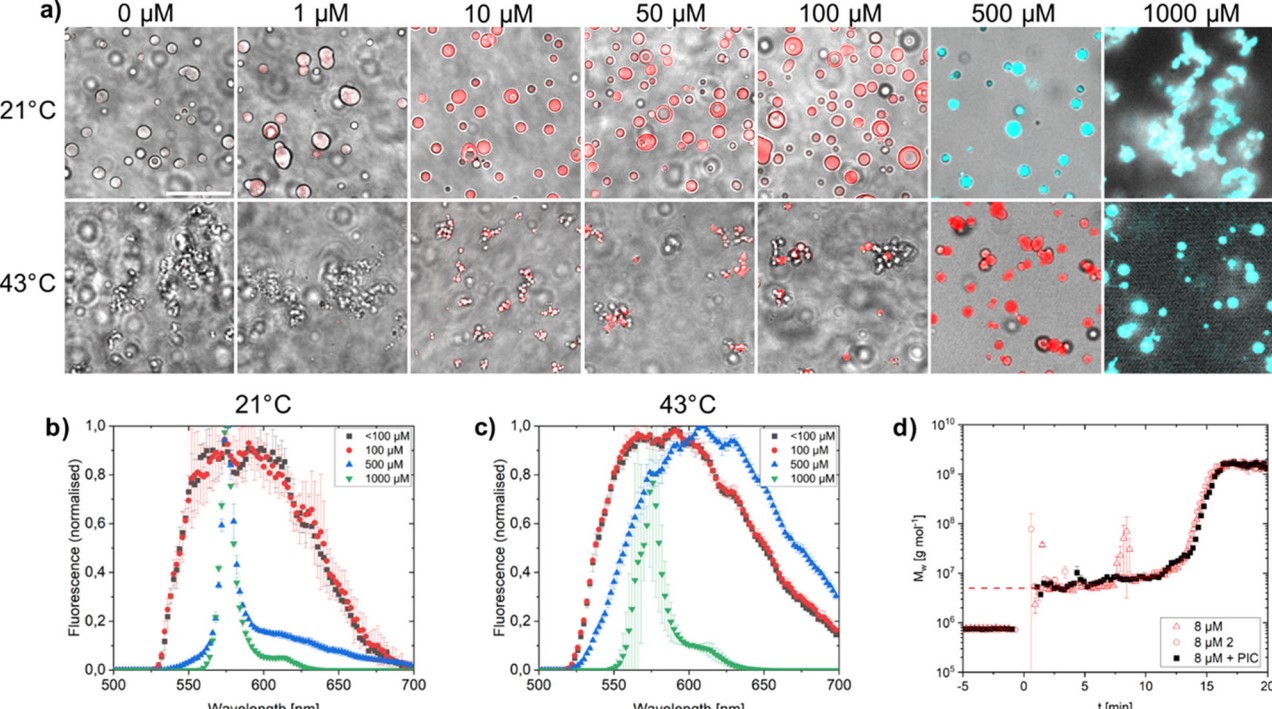

**Fig. 2 | PIC accumulation in FUS droplets.** **a** FUS droplets in PIC solution at various concentrations after 60 min incubation at 21 and 43 °C. PIC aggregates (blue) and monomers (red) were assigned based on their fluorescence intensity and spectrum (**b**, **c**). Scale bar: 20 μm. **d** Evolution of the weight-averaged molar mass $M_w$ of the droplet formation of 8 μM FUS after the addition of 0.1 g/l TEV without (**Δ,○**) and with (**■**) 50 μM PIC. The dashed red line represents the weight-averaged molar mass of pure FUS in buffer. Weight-averaged molar masses corresponding to negative time values represent TEV before the addition of FUS. The addition of FUS is $t = 0$.

**Fig. 3 | PIC aggregates are suppressed by over-crowding. a** cLSM images of J-aggregates formed at 5 mM concentration in the presence of Ficoll 70, accumulated on the microscopy slide surface at 21 °C. The aggregates dissolve at increasing Ficoll 70 (wt%) concentration. The scale bar represents 100 µm. **b** Threshold temperature for the formation of J-aggregates at 6 mM PIC in dependency of Ficoll 70 concentration. The threshold temperature increased up to 0.025 g/ml. Beyond this Ficoll content, the threshold temperature decreased. Solid red line refers to mean values of 2 technical replicates.

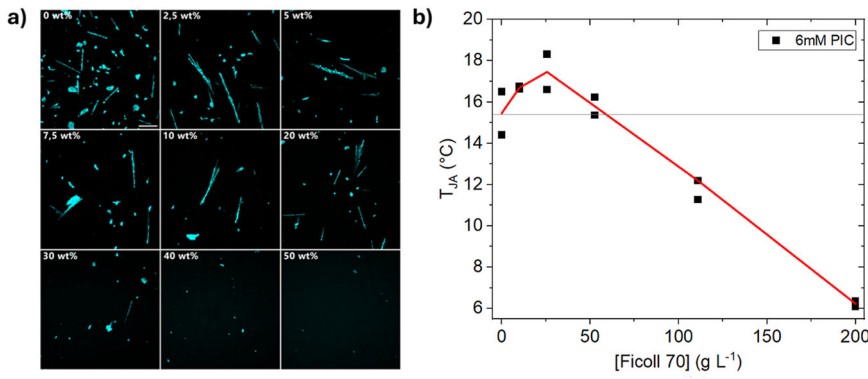

## Discussion

In summary, cellular stresses lead to a change in crowding favouring the monomeric compared to the aggregated state of PIC. This is unexpected since PIC aggregation is favoured by crowding and there is clear evidence that the crowding density increases under cell stress: the cell volume decreases[14,38,39] and the cytoplasm solidifies, possibly due to the association of heat-sensitive, unfolded proteins[15]. The diffusion of the intracellular biomolecules on different timescales is reduced[14,15].

Previous hyperosmotic stress experiments in mammalian cells have shown minor changes in the compaction and dynamics of IDP ProTα but pronounced changes in its translational diffusion[40]. The authors attributed such effects to a hindered diffusion on different lengths-scales, mimicked in vitro by a limited mesh size of concentrated PEG 35000 solutions[22]. Likewise, the increase in crowding beyond physiological levels (also referred to as an overcrowding condition[41]) could lead to a dissolution of intracellular aggregates due to a lack of space required to accommodate the large J-aggregates[42]. This disfavours the J-aggregated states compared to monomeric or oligomeric states for a similar reason as larger molecules are more frequently excluded from pores in size exclusion chromatography than smaller molecules are. Such a reversal of crowding effects at very high crowding density was previously observed for the *E. coli* enzyme dihydrofolate reductase[43]. We reproduced these findings with the crowding agent Ficoll 70, observing PIC disaggregation at concentrations where Ficoll polymer chains entangle to form a limited-sized mesh[34,36].

PIC aggregation was previously reported to cause a cytosolic accumulation of $Zn^{2+}$ and other ions ($Na^+$ and $K^+$)[20] and the threshold temperature for the formation of J-aggregates is responsive to these ions[21]. However, an increase in ion concentration would promote aggregation of PIC, the opposite effect to what was observed experimentally. The altered dynamics and morphologies of stress granules found here after PIC application resemble those resulting from the accumulation of misfolded proteins during heat stress or arsenate stress[3]. An imbalance in the $Zn^{2+}$ homeostasis is found under oxidative stress conditions[44]. Schelenker et al. reported a reduction of the cellular volume under oxidative stress on the same scale as for hyperosmotic shock[45]. Given the known intracellular increase in macromolecular crowding upon hyperosmotic stress[46], an eventual oxidative response to PIC application should result in increased crowding and consequently PIC disaggregation as well. In summary, the results suggest that it is cellular crowding that causes PIC to be monomeric under stress.

Considering the partitioning of the cytoplasm upon stress, it is remarkable that PIC is sequestered into stress granules but also does not aggregate in the condensed phase. This could indicate that the stress granules are at least as crowded as the cytoplasm. In fact, there is experimental evidence that condensates are commonly more crowded compared to their surrounding cellular fluids[41,47]. However, regarding stress granules and nuclear speckles, Kim et al. showed that the refractive indices are indistinguishable from the cytoplasm and nucleoplasm, respectively,

suggesting a similar crowding density[47]. In both cases, the highly crowded environments would support monomeric PIC. The experiments using FUS condensates in vitro support this hypothesis. FUS droplets are highly crowded, [1]H-NMR experiments of the low-complexity domain FUS revealed a minimum concentration of 477 mg/ml inside droplets[48], and other studies found a concentration of FUS-eGFP of 337.3 ± 8.2 mg/ml[49]. We found that the concentration threshold for aggregation of PIC inside the droplet is comparable to the dilute buffer solution. It is remarkable that also heat stress-treated FUS droplets, showing solidification by a decrease in circularity, do not promote PIC aggregation but rather repress it. Further, the experiments showed that PIC monomers partitioned into condensates while aggregates (when preformed in solution) were adsorbed on the FUS droplet surface. The pore-like cavities of FUS droplets are 2.8 nm in size (FUS-to-FUS)[50], efficiently excluding J-aggregates. In agreement with our experiments, Kim et al. showed that the fluorescent protein mCherry could still diffuse into stress granules while a mCherry tandem dimer protein showed significantly less partitioning[47]. Kamagata et al. observed a recruitment of client proteins into FUS droplets restricted to disordered chains that could be accommodated inside the droplets´ voids[51]. This volume exclusion effect is another hint that the highly crowded (over-crowded) conditions disfavour J-aggregate formation.

The principles of protein aggregation under stress conditions involve mechanisms beyond changes in crowding levels, including signalling pathways (e.g., PTMs)[52], pH changes[53], ATP depletion[54] and direct effects of stress on biomolecular conformational equilibrium[55,56]. The most well-known examples are heat- and oxidative-stress-induced protein unfolding, which expose hydrophobic patches of unfolded proteins that promote aggregation[56–58]. The emerging picture is that changes in crowding levels work together with other protective mechanisms to effectively respond to stress conditions. For example, increased crowding could enhance chaperone activity[59], thereby promoting faster remodelling of protein aggregates[60]. Another study showed that both increased crowding and high ionic strength promote the assembly of stress granules in response to osmotic shock[61]. Cell volume reduction and the resulting increase in crowding seem to be common responses to various stresses, including hyperosmotic[46], oxidative[38], heat[39] and starvation[62] conditions. In this context, our findings on PIC disaggregation suggest that cellular crowding plays a significant cytoprotective role under these stress conditions. The extent of changes in crowding levels and their potential functional consequences on the regulatory biochemical pathways involved in stress response require further studies.

In summary, our studies show that the change in crowding upon cell stress, induced by cell volume reduction, solidification of the cytoplasm or phase separation, does not promote PIC aggregation processes, the opposite is the case. Another important finding is that phase separation, as a response to stress, does not promote aggregation in the dilute or the condensed phase. It should be emphasized again that this study was designed to explicitly probe the environmental effects excluding other cellular processes under

heat stress like protein quality control mechanisms or post-translational modifications. It is further clear that the capacity of PIC to mimic the fibrillation of disease-related proteins is limited and further non-protein crowding sensors can expand the understanding of intracellular crowding effects on protein aggregation. However, this study suggests that increased crowding in the cytoplasm or in stress granules upon cellular stress is not an intrinsic cause for the aggregation of disease-related proteins in these respective environments.

## Methods

### Cell culture
HeLa cells were cultured in growth medium of Dulbecco's Modified Eagle's Medium (DMEM, Gibco, Carlsbad, USA) with 10% foetal bovine serum (FBS, Sigma Aldrich Chemie GmbH, Steinheim am Albuch, Germany) and 1% penicillin–streptomycin (Gibco, Carlsbad, USA) in a 37 °C incubator (Galaxy 170S, Eppendorf) at 10% $CO_2$ using T-25 cell culture flasks (Sarstedt AG & Co., Nümbrecht, Germany). The HeLa cell line expressing human FUS labelled at the C-terminus by GFP (HeLa FUS-GFP) was a gift from S. Alberti (TU Dresden, Germany). Cells were cultured using additional 400 μg/ml Geneticin (Thermo Fisher, Waltham, USA). At ~90% confluence cells were transferred to 10 mm fluorodishes (WPI, Friedberg, Germany) and incubated for at least 18 h.

### Heat stress and PIC treatment
To induce heat stress, HeLa cells in fluorodishes were washed twice with 200 μl Dulbecco's Phosphate-Buffered Saline (DPBS, from Gibco) and after applying new growth medium transferred into another incubator (Galaxy 170S, Eppendorf) maintaining 5% $CO_2$ at 43 °C. Cells were kept under these conditions for the time period of heat stress and then transferred into the incubation chamber (Okolab) maintaining 5% $CO_2$ and 43 °C. For measurements with PIC, 50 μM PIC was added to the medium before heat stress. Arsenate stress was induced using 1 mM sodium arsenate (Sigma Aldrich).

### Maltose binding protein (MBP)-FUS purification
For the expression of the recombinant MBP-FUS-6xHis, the plasmid vector pMal-TEV-Flag-FUS-TEV-6xHis was used, which was kindly provided by Prof. Dorothee Dormann (Institute of Molecular Cell Biology, Mainz)[63]. The vector was transformed into LB medium cultured NiCo21 (DE3) competent *E. coli*. Protein expression was induced by 100 μM IPTG at an OD of ~0.8 (600 nm) and purified using a nickel affinity column (Bio-Scale Mini Profinity IMAC cartridge, BIO-RAD) in an NGC BioRad chromatography system (~4 °C). To avoid phase separation, a high concentration of NaCl (300 mM) was used. The equilibration and elution buffers used for His-tag purification contained 50 mM $Na_2HPO_4$/$NaH_2PO_4$ pH = 8.0, 300 mM NaCl, 40 mM imidazole, 10 μM $ZnCl_2$, 4 mM βME and 50 mM $Na_2HPO_4$/$NaH_2PO_4$ pH = 8.0, 300 mM NaCl, 250 mM imidazole, 10 μM $ZnCl_2$, 4 mM βME, respectively. His-tag purified protein was given into an amylose column (NEW England BioLabs or NEB) pre-equilibrated with the equilibration buffer mentioned above and eluted in the next step with a solution of 50 mM $Na_2HPO_4$/$NaH_2PO_4$ pH = 8.0, 300 mM NaCl, 40 mM imidazole, 10 μM $ZnCl_2$, 4 mM βME and 20 mM maltose. This second purification step was performed at room temperature. At the end, the protein sample was dialyzed against $Na_2HPO_4$/$NaH_2PO_4$, pH = 8.1, 150 mM NaCl, 5% glycerol, 4 mM βME, 10 μM $ZnCl_2$ buffer solution and concentrated to ~40 μM by centrifugation in Amicon Filters.

### Preparation of MBP-FUS droplets in vitro
A solution of 10 μM FUS-MBP, 0.1 mM TEV and 0, 1, 10, 50, 100, 500 or 1000 μM PIC in buffer (20 mM Na–P-buffer; pH = 7, 150 mM NaCl, 2.5% glycerol, 1 mM DTT) was given into a low protein-binding tube up to a volume of 25 μl. After 1 h of incubation at 21 °C, or at 43 °C, FUS droplets accrued and were transferred into the incubation chamber (Okolab) maintaining 21 °C, respectively, 43 °C.

### Crowding studies using Ficoll 70
A stock solution of 50 wt% Ficoll in MiliQ water was prepared in a Falcon tube. Ficoll 70 was diluted to concentrations of 0, 2.5, 5, 7.5, 10, 20, 30, 40 and 50 wt% and supplemented with 5 mM PIC in a volume of 200 μl. The sample was incubated at RT for 1 day. In total, 25 μl were transferred to a protein low-binding 384-well plate and imaged using cLSM.

### Confocal laser scanning microscopy (cLSM)
Fluorescence imaging was performed using a confocal microscope (Olympus FV3000) with 488 nm excitation wavelength and the beam splitter DM405/488 for FUS-GFP and PIC. For imaging, a 60× silicone-oil immersion objective (UPLSAPO60XS2, NA = 1.3) was used. Emission was detected using GaAsP photomultiplier tubes. FUS-GFP was detected at 500–520 nm with no overlap fluorescence from PIC occurring in this regime of the spectrum. PIC aggregates were detected at 565–585 nm and PIC monomers at 600–620 nm. Samples were incubated using a closed chamber (Okolab) at 5% $CO_2$ and 37 °C or 43 °C. To quantitatively analyze the amount of microscopically visible PIC aggregates in cells we conducted a spectral analysis to discern the minimum intensity threshold for the detection of PIC-J-aggregates, distinct from the PIC monomer signal. The threshold was established at an intensity level of 400 (out of 4095) at a wavelength of 575 nm. This determination was based on iterative analyses aimed at identifying the lowest intensity level that could reliably capture the characteristic narrow peak at 575 nm, indicative of J-aggregates (Supplementary Fig. 11). In contrast, the monomeric form of PIC within the cytosol exhibited intensity levels ranging from 175 to 350, with an average of around 250, notably lacking the sharp peak at 575 nm. Spectral analyses focusing on samples exhibiting 400 intensity levels at 575 nm within the cytosol consistently revealed the unique signature associated with J-aggregates (Supplementary Fig. 11, inset). The threshold was not reduced further from 400, despite a standard deviation of around 30, to maintain a clear distinction between monomer and aggregate signals, thus preventing any overlap in their intensity levels. Afterwards, the nucleoli in the cell images were cut out to prevent artificial detection of bright monomer fluorescence in the nucleus. Via a semiautomatic script in CellProfiler 4.2.4 J-aggregates of PIC in the cytoplasm and co-localizations of PIC in stress granules were recorded.

### Fluorescence recovery after photobleaching (FRAP)
Stress granules enriched with FUS-GFP (PIC treated or untreated) were bleached with the 488 nm laser at 100% intensity and the recovery was detected at 500–520 nm using the 60× silicone-oil immersion objective (UPLSAPO60XS2, NA = 1.3). After background subtraction, normalization of the time-dependent mean intensity of the fluorescence was performed via Origin 2021b according to the easyFRAP protocol[64] and the mobile fraction was evaluated via the single exponential model

$$I(t)_{norm}^{fullscale} = I_0 - a * e^{-\beta*t} \tag{1}$$

$I_0$ refers to the mobile fraction of FUS-GFP and $\beta$ to the rate of recovery.

### Determination of the circularity, area size and mean fluorescence of droplets and stress granules
Droplet and cell images were evaluated via ImageJ 1.52a. Each fluorescence spot of PIC in droplets and stress granules was marked by using the "Wand tool" of ImageJ at the corresponding threshold level. In addition, 10 background spots were marked for later background subtraction. The "area", "shape description" and "mean grey value" dialogues were read out, saved as text-data and transferred to Origin 2021b. After background correction, the area size and fluorescence were displayed in boxplots. Significances were calculated using Origin's ANOVA analysis or two-tailed *t*-test.

## Fluorescence and absorption spectroscopy for in vitro experiments

To record fluorescence spectra of monomeric PIC, a 100 µM aqueous solution was used and for the J-aggregate an 8 mM solution. The solution was placed in a 3 × 3 mm quartz cuvette with a total volume of 50 µl. For the fluorescence spectra, the fluorescence spectrometer FP-8500 from Jasco was used. The settings of the measurements were: excitation wavelength at 488 nm, scanning speed of 200 nm/min and wavelength detection of 500–700 nm in 1 nm step size with excitation slit size of 5.0 nm and detection 10.0 nm. The effect of bovine serum albumin (BSA, Sigma Aldrich) and total RNA from yeast (Roche) and the droplet buffer were investigated under the same settings. For the absorption spectra, the absorption spectrometer V-770 from Jasco was used. The measurement settings were: a scanning speed of 200 nm/min and a wavelength detection of 400–600 nm at a 1 nm step size. For all experiments, a 3 × 3 mm quartz cuvette with a total volume of 50 µl was used with the exception of 1000 µM PIC, where the cuvette was exchanged for a 1 mm cuvette, due to strong absorption.

## PIC temperature-dependent aggregation experiments with UV–Vis spectroscopy

UV–Vis spectroscopy was carried out with a *Lambda-19* spectrometer (Perkin Elmer (Waltham, USA)) and a copper block sample holder, thermally equilibrated by an external thermostat. A demountable cuvette with a path length of 0.01 cm (Hellma Analytics (Mühlheim, Germany)) was used. To ensure absorptions below the maximum detector threshold, a slit width of 2 mm was applied. The aggregation temperature ($T_{JA}$) at a fixed concentration was determined by the appearance of a sharp peak at $A_{573}$ (J-aggregate) when gradually decreasing the temperature. The experiment started at 25 °C.

## Light scattering experiments

All scattering experiments were performed at $T = 25$ °C. Cylindrical light scattering cuvettes from Hellma (Mühlheim, Germany) with a diameter of 1 cm were used. To remove dust, the TEV solutions were filtered into scattering cells using Millex-GV (PVDF, 0.22 µm) syringe filters from Merck Millipore (Billerica, USA). Accordingly, a solution of 0.1 g/l TEV in buffer (20 mM Na–P-buffer, pH = 7, 150 mM NaCl, 2.5% glycerol, 1 mM DTT) was present in the cuvette. The solution was equilibrated in the light scattering device for 5 min at 25 °C. The scattering of the TEV in buffer was measured at least 5 min before MBP-FUS was added. A concentrated MBP-FUS solution in buffer was prepared at a concentration of 47.7 mg/l and added to the cuvette without filtration resulting in an MBP-FUS concentration of 8 mg/l thus initiating phase separation. The scattering intensity at each time point during the successive time-resolved analysis was recorded for 10 s/time frame.

Time-resolved light scattering experiments were carried out with the multi-detection laser light scattering system ALV/CGS-3/MD-8. A He–Ne laser with a wavelength of 632.8 nm was used as a light source. The system provides eight detectors positioned at angular increments of 8°, which allows simultaneous time-resolved SLS and DLS. An angular range of 30° ≤ θ ≤ 86° was measured, corresponding to a $q$-range in water of $6.8 \cdot 10^{-3} \leq q \leq 18 \cdot 10^{-3}$ nm$^{-1}$ with

$$q = \frac{4\pi n}{\lambda_0} \sin\left(\frac{\theta}{2}\right) \qquad (2)$$

the momentum transfer vector, $n$ the refractive index of the solvent, $\theta$ the scattering angle and $\lambda_0$ the laser wavelength in vacuum.

SLS provides the excess Rayleigh ratio $\Delta R_\theta$ of the solute

$$\Delta R_\theta = RR_{\theta,std}\left(\frac{r_{\theta,sol} - r_{\theta,solv}}{r_{\theta,std}}\right) \qquad (3)$$

with $RR_{\theta,std}$ corresponding to the absolute Rayleigh ratio of toluene used as standard and with $r_{\theta,solv}$, $r_{\theta,sol}$ and $r_{\theta,std}$ corresponding to the measured scattering intensity of the solvent, solution and the standard toluene, respectively. Before the addition of MBP-FUS, scattering intensities were evaluated using the Zimm equation[65]

$$\frac{Kc}{\Delta R_\theta} = \frac{1}{M_w} + \frac{R_g^2 q^2}{3 M_w} + 2A_2 c \qquad (4)$$

and after the addition of MBP-FUS the Guinier approximation[66] was used for evaluation the SLS data

$$\ln\left(\frac{Kc}{\Delta R_\theta}\right) = \ln\left(\frac{1}{M_w}\right) - \frac{R_g^2}{3} q^2 \qquad (5)$$

where $c$ is the concentration of the protein in g/l, $M_w$ is the weight average molecular mass in g/mol, $R_g^2$ is the $z$-averaged, squared radius of gyration in nm$^2$, $A_2$ is the second virial coefficient and $K$ is the contrast factor given by

$$K = \frac{4\pi^2 n_0^2}{N_A \lambda^4}\left(\frac{dn}{dc}\right)^2 \qquad (6)$$

with $n_0 = 1496$ the refractive index of the bath liquid (toluene) of the goniometer, $n = 1332$ the refractive index of the solvent, $N_A$ Avogadro's number, dn/dc = 0.187 ml/g the refractive index increment of proteins in water, and $\lambda = 632.8$ nm the wavelength of the laser. The concentration-dependent term $2A_2 c$ had to be neglected during the course of time-resolved experiments.

DLS measurements were recorded in the same $q$-regime as used for SLS. Evaluation of the resulting DLS data was done with the cumulant analysis of the electric field-time correlation function $g_1(\tau)$. An apparent diffusion coefficient $D$ is obtained from cumulant analysis[67] of $g_1(\tau)$.

$$\ln[g_1(\tau)] = A - q^2 D\tau + q^4 k_2 \tau^2 \qquad (7)$$

with $A$, $q^2 D$, and $k_2$ the zeroth, first, and second cumulant. A $z$-averaged diffusion coefficient, $D_z$, is determined by extrapolation of the resulting $D$ towards $q^2 = 0$ according to

$$D = D_z(1 + q^2 R_g^2 C + k_d c) \qquad (8)$$

where $C$ and $k_D$ are constants accounting for the angular and concentration dependency of the apparent diffusion coefficient. The concentration-dependent term $k_d c$ had to be neglected during the course of time-resolved experiment. $D_z$ is used to calculate the effective hydrodynamic radius $R_h$ using the Stokes–Einstein equation

$$R_h = \frac{k_b T}{6\pi\eta D_z} \qquad (9)$$

with $k_b$ the Boltzmann constant, $T$ the temperature in Kelvin and $\eta = 0.891$ mPa s the dynamic viscosity of the solvent.

The ratio of the radius of gyration and the hydrodynamic radius

$$\rho = \frac{R_g}{R_h} \qquad (10)$$

is a structure-sensitive parameter. Expected values for $\rho$ are 0.78 for spheres[68].

## Reporting summary

Further information on research design is available in the Nature Portfolio Reporting Summary linked to this article.

## Data availability

Raw files for the fluorescence spectra presented in Figs. 2b, c and Supplementary Figs. 2, 3, 4, 5a, 6, 9, 11 as well as absorption spectra presented in Supplementary Fig. 1 and 5b are provided in Supplementary Data 1. Raw data not presented in the manuscript are available upon request.

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

## Acknowledgements

We kindly acknowledge the funding of this project by the German Research Foundation (DFG-SPP 2191 #402723784, project #457267422 and the Cluster of Excellence RESOLV EXC 1069). We thank Rachida Salia Rachid Ussuale for experimental help and Simon Alberti and Dorothee Dormann for providing cells and FUS plasmids, respectively.

## Author contributions

R. Pollak and B. König performed the experiments and data evaluation. L. Koch performed and evaluated the light scattering experiments. K. Huber and S. Ebbinghaus designed the project. S. S. Ribeiro and N. Samanta purified the MBP-FUS protein. All authors contributed to the discussion and analysis of the results and prepared the manuscript.

## Funding

## Competing interests

The authors declare no competing interests.
