## [Peer Review file · Communications Chemistry]

Cell stress and phase separation stabilize the monomeric state of pseudoisocyanine chloride employed as a self-assembly crowding sensor

Corresponding Author: Professor Simon Ebbinghaus

Version 0:

Reviewer comments:

Reviewer #1

(Remarks to the Author)

In this manuscript, Pollak and co-authors investigated whether crowding environment can induce aggregation of PIC, whose fluorescence spectrum is shifted and fluorescence intensity is enhanced upon aggregation. The authors observed that cellular stresses lead to a change in the physicochemical properties of the cellular milieu that favour the monomeric compared to the aggregated form of PIC. The observation reported here is interesting. In addition, I found the heterogenous distribution of PIC aggregates in preformed FUS droplets reported in Figure S5 interesting and the explanation to it convincing. I am unsure, however, how much would the observation made with PIC translate to the aggregation of disease-causing proteins. I agree that the PIC is a system that allows one to study aggregation without the confounding effect of PQC mechanisms. However, as the authors mentioned themselves, the capacity of PIC to mimic the fibrillation of disease-related proteins is limited.

Major Concerns:

1. Abstract: line 24-26: "The study provides strong evidence that the high crowding density in the cytoplasm and in SGs upon cellular stress may not be the cause of aggregation of disease-related proteins". As mentioned above, I am not sure how much of the observation made here with a small molecule can translate to proteins.
2. The paper discusses the effect of "crowding environment induced by stress", but did not characterize the crowdedness of the cellular environment before and after stress. Although literatures are cited, since this is the major parameter that is been investigated in the study, I think it would be better to confirm, experimentally, that the cellular environment got more crowded after stress, especially when the authors used a different cell line than the cited paper.
3. Line 148, and 159, "cellular stresses lead to a change in the physicochemical properties of the cellular milieu". This statement appears a few times, but it is unclear that are the physicochemical properties that has changed upon cellular stress. Does it refer to the FRAP experiment in Fig. 1C? That is a measurement of the stress granule. I am wondering what physicochemical properties changed in the cytoplasm that made PIC less prone to aggregates.
4. Figure S11, it is unclear how is the threshold temperature determined.
5. Line 64: "An intriguing question that we address in this work is whether the physicochemical changes of the cytoplasm associated with cell stress are a cause for protein aggregation". I feel that this is not an accurate description of the work, since PIC is not a protein.
6. Line 80: it would be nice to show "reversibility" experimentally. Figure S2 shows the fluorescence spectrum of the monomer in (a), and aggregates in (b). This figure does not show the transition between the two states and also does not show reversibility.
7. The manuscript has two main figures for the result section. I feel that Figure S10 and S11, which is mentioned in the discussion section, could be move into the main figures and result section.
8. The mechanism of why PIC show aggregation in stressed cells needs further investigation. The authors used Ficoll 70 to mimic the crowding environment of cell. To better connect this result with the results from cellular experiment, it would be helpful to measure the crowdedness of Ficoll 70 at difference concentration and compare that with the cellular environment.

Minor comments:

9. Line 298: missing figure number
10. Why does fluorescence spectrum in Figure 2C look different than Figure S2? For monomer, in Figure 2C, it is flat between 550nm-650nm, while in Figure S2, it peaks at ~570nm.
11. Please specify in the figure legend, what does the error bar stands for, how many repeats are in the experiment, and whether they are technical repeats or biological repeats.

Reviewer #2

(Remarks to the Author)

The manuscript entitled “Crowding under Cell Stress and Phase Separation mitigates Aggregation” by Pollak et al. presents a study of the application of stress to life cells. The cytosol is a highly crowded environment, but contrary to expectations the high concentration does not favour aggregation of unfolded proteins inside the cells. The study, conducted with various interesting techniques, is well described and the conclusions are sound. The novelty of the investigation is high. I am therefore recommending publication of the study after some corrections:

- 1) One of the keywords is liquid-liquid phase separation. However, LLPS is hardly mentioned in the manuscript, so it should or be better explained in this context or be replaced by another keyword.
- 2) The authors should mention that molecular diffusion dynamics can be perfectly studied by neutron scattering, even on life cells (p. 2).
- 3) The authors underline that crowding is not at the origin of protein aggregation in the cytosol. So could they explain or at least guess what is really the reason of aggregation or amyloid fibril formation under stress conditions ?
- 4) The excessive use of acronyms (PIC, FUS-GFP, HS, SG, etc.) makes it laborious to read the manuscript. The authors must reduce the number of abbreviations to a minimum to make the lecture more fluid.
- 5) The authors should further detail the protective effects of cellular crowding. Can they cite protective mechanisms against other stresses ?
- 6) Equations have to be numbered.
- 7) The number of self-citations is too high and must be reduced.

Reviewer #3

(Remarks to the Author)

The paper “Crowding under Cell Stress and Phase Separation mitigates Aggregation” centres around the behavior of the dye, pseudoisocyanine chloride (PIC), in HeLa cells under heat stress. The main claims of the paper rely on PIC being a good reporter of protein aggregation. This brings me to my major concern with the manuscript – I don't see any backup in the manuscript (or the cited references 10 and 11) that supports changes in PIC fluorescence reporting on anything other than PIC aggregation. And I don't see why PIC aggregation is of interest. Just because PIC aggregates doesn't mean that proteins are also aggregating. The title of the article doesn't make clear whether the authors think they are studying protein aggregation or PIC aggregation, but I think most readers would be hoping for protein aggregation.

In order to further consider this article further, I would need to see either evidence that PIC aggregation does indeed report on protein aggregation or some other interesting protein behavior, or see some justification for why PIC aggregation in cells is worthy of study.

Version 1:

Reviewer comments:

Reviewer #1

(Remarks to the Author)

The authors have sufficiently addressed my concerns in this revised manuscript.

Reviewer #2

(Remarks to the Author)

The authors answered in a satisfactory way to all my requests. I am thus recommending publication.

We would like to thank the reviewers for their valuable comments and suggestions, which helped us to significantly improve the manuscript by additional experiments and analysis. In the following text, we address the comments in a point-by-point response and mark any changes in the main manuscript in yellow.

Reviewers' comments:

Reviewer #1 (Remarks to the Author):

In this manuscript, Pollak and co-authors investigated whether crowding environment can induce aggregation of PIC, whose fluorescence spectrum is shifted and fluorescence intensity is enhanced upon aggregation. The authors observed that cellular stresses lead to a change in the physicochemical properties of the cellular milieu that favour the monomeric compared to the aggregated form of PIC. The observation reported here is interesting. In addition, I found the heterogenous distribution of PIC aggregates in preformed FUS droplets reported in Figure S5 interesting and the explanation to it convincing. I am unsure, however, how much would the observation made with PIC translate to the aggregation of disease-causing proteins. I agree that the PIC is a system that allows one to study aggregation without the confounding effect of PQC mechanisms. However, as the authors mentioned themselves, the capacity of PIC to mimic the fibrillation of disease-related proteins is limited.

Major

Concerns:

1. Abstract: line 24-26: “The study provides strong evidence that the high crowding density in the cytoplasm and in SGs upon cellular stress may not be the cause of aggregation of disease-related proteins”. As mentioned above, I am not sure how much of the observation made here with a small molecule can translate to proteins.

We thank reviewer 1 and 3 for raising this important point which helped us to clarify the aim and scope of this study. PIC features two highly interesting properties with respect to self-assembly processes of proteins in cells. First and foremost, PIC forms fibrillar aggregates via self-assembly^{1,2} like many proteins in living systems (e.g. actin, tubulin, fibrinogen and beta-amyloid³⁻⁸). Second, PIC in its monomeric state exhibits photophysical properties which significantly differ from the corresponding photo-physical properties in its self-assembled state. This suggests the use of PIC in two different research strategies:

(i) Crowding sensor for self-assembly processes

It is well known that cells are crowded and that the crowding density in terms of the dry mass per volume varies across a cell and between different cells. However, the effect of crowding on biomolecular processes in health and disease conditions is still unclear. The reason is that crowding effects are multi-faceted including excluded volume effects, soft interactions and solvent effects⁹⁻¹². They strongly depend on the size of the biomolecular system and its chemical composition (e.g. charge). For complex processes like protein aggregation crowding effects are versatile for different proteins, since aggregation occurs on different length scale with specific chemistry. Crowding can inhibit but also accelerate aggregation¹³. It becomes important to distinguish for example the crowding effect on the conformation of the monomer from the crowding effect on fibril formation. Cellular crowding sensors for monomeric states were developed and applied consisting of a disordered polyethylene glycol macromolecule¹⁴ or an artificial protein with two linked α -helices and fluorescent proteins¹⁵. Such studies revealed important insights into the change in dimensions of a monomeric biomolecule under different cellular conditions such as osmotic pressure or cell stress¹⁶⁻²¹. The crowding effect on self-assembly processes is however expected to be different, since the e.g. the change in excluded volume for the association of two monomers is larger compared to a conformational transition of the monomer itself²². Our previous work^{1,2}, including the comparison to other dyestuff²³, showed that PIC is a

well-suited self-assembly sensor which is applied here to characterize crowding effects on the self-assembly processes under heat stress and phase separation.

(ii) **Modell system to investigate fibrillation processes**

PIC maybe used as a model system to study fibrillation processes. The folding and aggregation pathway of disease-related proteins is very complex involving unfolded and misfolded structures, oligomers and fibrils. Cellular processing by PTM, PQC, cellular localization etc. further complicates the analysis. To address specific aspects of the pathway, for example truncated versions of proteins are investigated *in vitro* using extrinsic probes that are sensitive to fibrillization processes, e.g. ThT dyes²⁴. As such a model system, PIC could mimic a specific aspect of the aggregation reaction, fibrillization. However, this approach needs validation with mechanistic studies of disease related proteins to identify systems where PIC can serve as a model.

Our approach is based on strategy (i). We apply a specific crowding sensor (for self-assembly) and to monitor the change of the crowding effect under different cellular conditions. We assume that similar crowding effects govern the fibrillization of proteins. This is however only an assumption that needs to be validated by further experiments in strategy (ii).

Acknowledging the reviewers comments we changed the manuscript to clarify our aim and the scope of the manuscript.

According to the suggestion by reviewer 3 we changed the title from “Crowding under Cell Stress and Phase Separation mitigates Aggregation” to “Cell stress and phase separation stabilize the monomeric state of pseudoisocyanine chloride employed as a self-assembly crowding sensor”.

We rewrote the abstract: “Cellular stress and ageing involve an increase in crowding and aggregation of amylogenic proteins. We here investigate if crowding is the intrinsic cause of aggregation and utilise a previously established non-protein aggregation sensor, namely pseudoisocyanine chloride (PIC). PIC shows fibrillization in cells into a highly fluorescent J-aggregated state and is sensitive to crowding. Surprisingly, cell stress conditions stabilise the monomeric rather than the aggregated state of PIC both in the cytoplasm and in stress granules. Regarding the different physiochemical changes of the cytoplasm occurring upon cell stress, involving volume reduction, phase separation and solidification, the intrinsic crowding effect is not the key factor to drive associated self-assembly processes.”

We further added and modified the text throughout the manuscript to clarify the strategy and implications of the results (introduction and discussion sections) as well as a comparison to protein systems (discussion section).

2. The paper discusses the effect of “crowding environment induced by stress”, but did not characterize the crowdedness of the cellular environment before and after stress. Although literatures are cited, since this is the major parameter that is been investigated in the study, I think it would be better to confirm, experimentally, that the cellular environment got more crowded after stress, especially when the authors used a different cell line than the cited paper.

We also thank the reviewer for this important comment. As discussed in response to point 1, crowding effects are multi-faceted and the PIC sensor provides a unique read-out. Other sensors, like the above mentioned sensors by Gnutt and Boersma^{14,19}, report on monomer dimensions and the results will not be comparable. However, we indeed think that using these sensors is a great idea to elaborate on the crowding effects on protein stability upon cellular stress, but this would be a new comprehensive study.

We added this idea to the conclusion as an outlook (Line 275-277). We also emphasize in the discussion that increased cellular crowding is a common feature among several stresses and that further studies are required to investigate the extent of changes in crowding upon stress: Page 13 (lines 260-266): "Cell volume reduction and the resulting increase in crowding seem to be common responses to various stresses, including hyperosmotic, oxidative, heat and starvation conditions. In this context, our findings on PIC disaggregation suggest that cellular crowding plays a significant cytoprotective role under these stress conditions. The extent of changes in crowding levels and their potential functional consequences on the regulatory biochemical pathways involved in stress response require further studies."

3. Line 148, and 159, "cellular stresses lead to a change in the physicochemical properties of the cellular milieu". This statement appears a few times, but it is unclear that are the physicochemical properties that has changed upon cellular stress. Does it refer to the FRAP experiment in Fig. 1C? That is a measurement of the stress granule. I am wondering what physicochemical properties changed in the cytoplasm that made PIC less prone to aggregates.

We refer to changes such as cell volume reduction, phase separation or solidification under stress modifying the crowding effect measured by PIC. We made the text clearer by now referring directly to the crowding effect: "In summary, cellular stresses lead to a change in crowding favoring the monomeric compared to the aggregated form of PIC."

or giving further explanation:

"Our studies show that the change in crowding upon cell stress, induced by cell volume reduction, solidification of the cytoplasm or phase separation, does not promote PIC aggregation processes, the opposite is the case."

We also adapted this in the other examples.

4. Figure S11, it is unclear how is the threshold temperature determined.

We add the details for the experiment and calculation of the threshold temperatures in the materials and methods section (Lines 394-402).

5. Line 64: "An intriguing question that we address in this work is whether the physicochemical changes of the cytoplasm associated with cell stress are a cause for protein aggregation". I feel that this is not an accurate description of the work, since PIC is not a protein.

The sentence was changed to:

"An intriguing question that we address in this work is whether changes in the crowding effect of the cytoplasm associated with cell stress could promote self-assembly processes..." (Line 69)

6. Line 80: it would be nice to show "reversibility" experimentally. Figure S2 shows the fluorescence spectrum of the monomer in (a), and aggregates in (b). This figure does not show the transition between the two states and also does not show reversibility.

We conducted further experiments to show reversibility of the PIC aggregation in the droplet buffer to lower the aggregation concentration to 100 μ M PIC (Figure S5) with an increasing temperature measurement, followed by the cooling down of the same solution. In this PIC concentration the J-aggregate shifts with increasing temperature into a pure monomeric form

and reforms the J-aggregates with cooling down of the solution. The results are shown below revealing that the aggregation of PIC is a reversible process.

The experiments were added to the manuscript as SI Figure S3.

Figure 1. Temperature-dependent fluorescence spectra of PIC in droplet buffer. a) fluorescence peak of the J-aggregate at 20°C transitioning into the monomeric PIC fluorescence with rising temperature (20 to 45°C). b) The same solution was then cooled down, resulting in the reformation of the J-aggregate.

7. The manuscript has two main figures for the result section. I feel that Figure S10 and S11, which is mentioned in the discussion section, could be move into the main figures and result section.

We added Figure S10 and S11 as the new Figure 3 in the manuscript including a sub-chapter addressing further details to Ficoll 70 crowding (Page 9 Line 176).

8. The mechanism of why PIC show aggregation in stressed cells needs further investigation. The authors used Ficoll 70 to mimic the crowding environment of cell. To better connect this result with the results from cellular experiment, it would be helpful to measure the crowdedness of Ficoll 70 at difference concentration and compare that with the cellular environment.

We thank the reviewer for the remark comparing Ficoll 70 to cytosolic crowding conditions. Similar to the answer to point 2, to extend the study to characterizing Ficoll 70 crowding effects with other methods or sensors are beyond the scope of this work and could be even confusing. We use Ficoll 70 itself as a previously well-characterized benchmark for the PIC sensor. We addressed the reviewers suggestion by adding more details on the characterization of Ficoll70 to the manuscript on page 9 under section “Overcrowding suppresses PIC aggregate formation”.

“We tested the effect of increasing crowding conditions using Ficoll 70 as a crowding agent that can be studied to a crowding fraction up to ~50 wt%. Further it is a branched and internally cross-linked polysucrose that is a copolymer of sucrose and epichlorohydrin, which is often approximated as a prototypical excluded-volume crowder. Indeed, with an increased amount of crowding, the number of micrometre-sized aggregates decreased beyond 10 wt% leading to an almost complete repression at 50 wt% (Figure 3a). Further, the threshold temperature for the formation of PIC J-aggregates increased with the concentration of Ficoll 70 up to 2.5 wt% but then dropped and was even lower than that of pure PIC solution at 22.5 wt% (Figure 3b). This shows parallels with the increase of the excluded volume or reduction of physical space in Ficoll 70 solutions up to 2.5 wt%. However, at concentrations of > 2.5 wt% it is assumed that Ficoll 70 molecules undergo compression due to (over-)crowding and further at

10-15 wt% reaching an overlap concentration, where individual Ficoll 70 molecules are forming a polymer mesh inside the solution. This change in crowding environment is followed by stabilizing and destabilizing effects on the crowding sensor PIC as well as protein self-assembly.”

Minor comments:

9. Line 298: missing figure number

The figure number was added.

10. Why does fluorescence spectrum in Figure 2C look different than Figure S2? For monomer, in Figure 2C, it is flat between 550nm-650nm, while in Figure S2, it peaks at ~570nm.

These spectra were recorded under different conditions, PIC fluorescence in water (100 μ M, Fig. S2) shows pure monomer fluorescence with a narrow peak around ~560 nm. In the FUS-droplets (Fig. 2) the spectrum changes due to protein interactions leading to an intensity increase and broadening of the spectrum (compare Figure S4). The formation of multimers (fluorescence peak at 610 nm) in FUS could further contribute to the spectral broadening.

11. Please specify in the figure legend, what does the error bar stands for, how many repeats are in the experiment, and whether they are technical repeats or biological repeats.

Figure 1 includes the measurements summarized in Table S1 and S2. Further we added the information that the measurements were conducted at 5 independent days including 10 samples from a single cell batch. Every cell was measured as listed in Table S1. Therefore there are 346 cells as biological repeats.

Reviewer #2 (Remarks to the Author):

The manuscript entitled “Crowding under Cell Stress and Phase Separation mitigates Aggregation” by Pollak et al. presents a study of the application of stress to life cells. The cytosol is a highly crowded environment, but contrary to expectations the high concentration does not favour aggregation of unfolded proteins inside the cells. The study, conducted with various interesting techniques, is well described and the conclusions are sound. The novelty of the investigation is high. I am therefore recommending publication of the study after some corrections:

1) One of the keywords is liquid-liquid phase separation. However, LLPS is hardly mentioned in the manuscript, so it should or be better explained in this context or be replaced by another keyword.

Indeed, this was confusing. Liquid-liquid is now removed, since strictly speaking this is not the adequate terminology. Explanation is also added to the text (Line 57).

2) The authors should mention that molecular diffusion dynamics can be perfectly studied by neutron scattering, even on life cells (p. 2).

We included this idea on page 2 (Line 49) adding the sentence: “a slow-down of molecular diffusion dynamics as measured by single particle tracking experiments as well as neutron scattering”.

3) The authors underline that crowding is not at the origin of protein aggregation in the

cytosol. So could they explain or at least guess what is really the reason of aggregation or amyloid fibril formation under stress conditions?

We thank the reviewer for the helpful discussion. We addressed this question in two ways:
1st) We included two sentences in the introduction discussing how the protein quality control machinery (e.g. Hsp70) regulates protein folding and aggregation under stress conditions:

Page 2 (Lines 36-40): "An example is the heat shock protein 70 KDa (Hsp/Hsc70) family of chaperones that shares functions in protein folding, disaggregation, translocation and degradation. Overexpression of Hsp70 mitigates protein aggregation and enhances cell survival whereas lowered expression levels trigger protein aggregation and promote cell death⁵."

2nd) We added a paragraph in discussion section addressing the different factors and mechanisms leading to protein aggregation during stress conditions: Pages 12 and 13 (Lines 250-266): "The principles of protein aggregation under stress conditions involve mechanisms beyond changes in crowding levels, including signalling pathways (e.g., PTMs), pH changes, ATP depletion, and direct effects of stress on biomolecular conformational equilibrium. The most well-known examples are heat- and oxidative-stress-induced protein unfolding, which expose hydrophobic patches of unfolded proteins that promote aggregation. The emerging picture is that changes in crowding levels work together with other protective mechanisms to effectively respond to stress conditions. For example, increased crowding could enhance chaperone activity, thereby promoting faster remodelling of protein aggregates. Another study showed that both increased crowding and high ionic strength promote the assembly of stress granules in response to osmotic shock. Cell volume reduction and the resulting increase in crowding seem to be common responses to various stresses, including hyperosmotic, oxidative, heat and starvation conditions. In this context, our findings on PIC disaggregation suggest that cellular crowding plays a significant cytoprotective role under these stress conditions. The extent of changes in crowding levels and their potential functional consequences on the regulatory biochemical pathways involved in stress response require further studies."

4) The excessive use of acronyms (PIC, FUS-GFP, HS, SG, etc.) makes it laborious to read the manuscript. The authors must reduce the number of abbreviations to a minimum to make the lecture more fluid.

Done.

5) The authors should further detail the protective effects of cellular crowding. Can they cite protective mechanisms against other stresses?

Please see reply to point 3. We added details on protein quality control mechanisms, namely molecular chaperones and their role in preventing protein aggregation (1st point in reply to point 3) as well as examples where of cellular crowding has a protective effect (2nd point to reply in point 3) on stress response.

6) Equations have to be numbered.

The numbering was added to the equations.

7) The number of self-citations is too high and must be reduced.

We replaced or removed self-citations whenever possible, further references added in the new discussions are not from our own work.

Reviewer #3 (Remarks to the Author):

The paper “Crowding under Cell Stress and Phase Separation mitigates Aggregation” centres around the behavior of the dye, pseudoisocyanine chloride (PIC), in HeLa cells under heat stress. The main claims of the paper rely on PIC being a good reporter of protein aggregation. This brings me to my major concern with the manuscript – I don’t see any backup in the manuscript (or the cited references 10 and 11) that supports changes in PIC fluorescence reporting on anything other than PIC aggregation. And I don’t see why PIC aggregation is of interest. Just because PIC aggregates doesn’t mean that proteins are also aggregating. The title of the article doesn’t make clear whether the authors think they are studying protein aggregation or PIC aggregation, but I think most readers would be hoping for protein aggregation.

In order to further consider this article further, I would need to see either evidence that PIC aggregation does indeed report on protein aggregation or some other interesting protein behavior, or see some justification for why PIC aggregation in cells is worthy of study.

We thank the reviewer for this important and helpful discussion, in line with reviewer 1. We ask the reviewer to please refer to the answer to reviewer 1 point 1.

References

1. Hämisch, B., Pollak, R., Ebbinghaus, S. & Huber, K. Self-Assembly of Pseudo-Isocyanine Chloride as a Sensor for Macromolecular Crowding In Vitro and In Vivo. *Chem. Eur. J.* **26**, 7041–7050 (2020).
2. Hämisch, B., Pollak, R., Ebbinghaus, S. & Huber, K. Thermodynamic Analysis of the Self-Assembly of Pseudo Isocyanine Chloride in the Presence of Crowding Agents. *ChemSystemsChem* syst.202000051 (2021) doi:10.1002/syst.202000051.
3. Freeman, R. *et al.* Reversible self-assembly of superstructured networks. *Science* **362**, 808–813 (2018).
4. Chafai, D. E. *et al.* Reversible and Irreversible Modulation of Tubulin Self-Assembly by Intense Nanosecond Pulsed Electric Fields. *Advanced Materials* **31**, 1903636 (2019).
5. Saha, S., Büngeler, A., Hense, D., Strube, O. I. & Huber, K. On the Mechanism of Self-Assembly of Fibrinogen in Thrombin-free Aqueous Solution. *Langmuir* **40**, 4152–4163 (2024).
6. Portillo, A. *et al.* Role of monomer arrangement in the amyloid self-assembly. *Biochimica et Biophysica Acta (BBA) - Proteins and Proteomics* **1854**, 218–228 (2015).
7. Pignataro, M. F., Herrera, M. G. & Doderio, V. I. Evaluation of Peptide/Protein Self-Assembly and Aggregation by Spectroscopic Methods. *Molecules* **25**, 4854 (2020).
8. Jansen, R., Dzwolak, W. & Winter, R. Amyloidogenic Self-Assembly of Insulin Aggregates Probed by High Resolution Atomic Force Microscopy. *Biophysical Journal* **88**, 1344–1353 (2005).
9. Gnutt, D. & Ebbinghaus, S. The macromolecular crowding effect -- from in vitro into the cell. *Biological Chemistry* **397**, 37–44 (2016).
10. Miklos, A. C., Li, C., Sharaf, N. G. & Pielak, G. J. Volume Exclusion and Soft Interaction Effects on Protein Stability under Crowded Conditions. *Biochemistry* **49**, 6984–6991 (2010).
11. Ellis, R. J. Macromolecular crowding: an important but neglected aspect of the intracellular environment. *Curr. Opin. Struct. Biol.* **11**, 114–119 (2001).
12. Das, N. & Sen, P. Shape-Dependent Macromolecular Crowding on the Thermodynamics and Microsecond Conformational Dynamics of Protein Unfolding Revealed at the Single-Molecule Level. *J. Phys. Chem. B* **124**, 5858–5871 (2020).

13. Gao, M. *et al.* Crowders and Cosolvents-Major Contributors to the Cellular Milieu and Efficient Means to Counteract Environmental Stresses. *Chemphyschem* **18**, 2951–2972 (2017).
14. Gnutt, D., Gao, M., Brylski, O., Heyden, M. & Ebbinghaus, S. Excluded-Volume Effects in Living Cells. *Angew. Chem. Int. Ed.* **54**, 2548–2551 (2015).
15. Liu, B. *et al.* Design and Properties of Genetically Encoded Probes for Sensing Macromolecular Crowding. *Biophysical Journal* **112**, 1929–1939 (2017).
16. Liu, B., Hasrat, Z., Poolman, B. & Boersma, A. J. Decreased Effective Macromolecular Crowding in Escherichia coli Adapted to Hyperosmotic Stress. *J Bacteriol* **201**, (2019).
17. Taloni, A. *et al.* Volume Changes During Active Shape Fluctuations in Cells. *Phys. Rev. Lett.* **114**, 208101 (2015).
18. Liu, B. *et al.* Influence of Fluorescent Protein Maturation on FRET Measurements in Living Cells. *ACS Sens.* **3**, 1735–1742 (2018).
19. Liu, B., Poolman, B. & Boersma, A. J. Ionic Strength Sensing in Living Cells. *ACS Chem. Biol.* **12**, 2510–2514 (2017).
20. Gnutt, D., Brylski, O., Edengeiser, E., Havenith, M. & Ebbinghaus, S. Imperfect crowding adaptation of mammalian cells towards osmotic stress and its modulation by osmolytes. *Mol. BioSyst.* **13**, 2218–2221 (2017).
21. Gnutt, D., Sistemich, L. & Ebbinghaus, S. Protein Folding Modulation in Cells Subject to Differentiation and Stress. *Front. Mol. Biosci.* **6**, 38 (2019).
22. Zhou, H.-X., Rivas, G. & Minton, A. P. Macromolecular Crowding and Confinement: Biochemical, Biophysical, and Potential Physiological Consequences *. *Annu. Rev. Biophys.* **37**, 375–397 (2008).
23. Koch, L., Pollak, R., Ebbinghaus, S. & Huber, K. A Comparative Study on Cyanine Dye-stuffs as Sensor Candidates for Macromolecular Crowding In Vitro and In Vivo. *Biosensors* **13**, 720 (2023).
24. Vöpel, T. *et al.* Inhibition of Huntingtin Exon-1 Aggregation by the Molecular Tweezer CLR01. *J. Am. Chem. Soc.* **139**, 5640–5643 (2017).